# Health status of individuals referred to first-line intervention for hip and knee osteoarthritis compared with the general population: an observational register-based study

Kristin Gustafsson [1,2] Joanna Kvist [1] Marit Eriksson [3]
Andrea Dell'Isola [4] Caddie Zhou,[5] Leif E. Dahlberg [4] Ola Rolfson [6]

For numbered affiliations see end of article.

**Correspondence to**
Mrs Kristin Gustafsson;
kristin.gustafsson@rjl.se

## ABSTRACT

**Objectives** To describe the prevalence of comorbidities in a population referred to standardised first-line intervention (patient education and exercise) for hip and knee osteoarthritis (OA), in comparison with the general population. Furthermore, we aimed to evaluate if eventual differences were associated with socioeconomic inequalities.

**Design** Register-based study.

**Setting** Primary healthcare, Sweden.

**Participants** Individuals with hip and/or knee OA included in the Better Management for Patients with Osteoarthritis Register between 2008 and 2016 and and an age-matched, sex-matched and residence-matched reference cohort (1:3) from the general Swedish population.

**Outcome measures** Comorbidities were identified with the RxRisk Index, the Elixhauser Comorbidity Index and the Charlson Comorbidity Index, and presented with descriptive statistics as (1) individual diseases, (2) disease categories and (3) scores for each index. The prevalence of comorbidities in the two populations was tested using logistic regression, with separate analyses for age groups and the most affected joint. We then adjusted the analyses for socioeconomic status.

**Results** In this OA population, 85% had ≥1 comorbidity compared with 78% of the reference cohort (OR; 1.62 (95% CI 1.59 to 1.66)). Cardiovascular/blood diseases were the most common comorbidities in both populations (OA, 59%; reference, 54%), with OR; 1.22 (95% CI 1.20 to 1.24) for the OA population. Younger individuals with OA were more comorbid than their matched references overall, and population differences decreased with age (eg, ≥3 comorbidities, aged ≤45 years OR; 1.74 (95% CI 1.52 to 1.98), ≥81 years OR; 0.95 (95% CI 0.87 to 1.04)). Individuals with knee OA were more comorbid than those with hip OA overall. Adjustment for socioeconomic status did not change the estimates.

**Conclusion** Comorbidities were more common among individuals with hip and knee OA than among matched references from the general population. The differences could not be explained by socioeconomic status.

**Trial registration number** NCT03438630.

## Strengths and limitations of this study

► This study describes comorbidities in a nationwide osteoarthritis (OA) population in the 3 years before participation in a standardised first-line intervention for hip and knee OA, compared with matched individuals from the general population.

► The study captures data on comorbidities and socioeconomic status from objective data sources, not self-reported data, which gives a good opportunity to adjust for socioeconomic differences.

► Comorbidities are identified with diagnostic codes (International Classification of Diseases, 10th revision (ICD-10) codes) and prescription of drugs (Anatomical Therapeutic Chemical (ATC) codes) using three comorbidity indices: the RxRisk Index, the Elixhauser Comorbidity Index and the Charlson Comorbidity Index, and are also described as individual diseases and as disease categories.

► To determine if socioeconomic inequalities could explain differences in comorbidity between the populations, the results were adjusted for highest educational level, country of birth and marital status.

► In this study we were only able to capture ICD-10 codes from healthcare delivered through inpatient care and specialised outpatient visits; the ATC codes included data on drugs prescribed in primary healthcare, in hospitals, by private caregivers and occupational healthcare.

## INTRODUCTION

Osteoarthritis (OA) is one of the most common musculoskeletal disorder in older adults.[1] Approximately 25% of the population more than 45 years old have OA, without accounting for the large numbers who never seek healthcare for their problem.[2] OA develops slowly over time, leading to pain, joint stiffness and functional limitations.[3]

Coexistence of comorbidities is common among individuals with OA. More than

65% have at least one other chronic condition, which increases with age.[4 5] The most frequent pattern of OA comorbidities includes cardiovascular and/or metabolic diseases[6] as well as psychological comorbidities such as depressive symptoms.[7 8] The higher frequency of specific comorbidities in people with OA can be partially explained by an increased vulnerability to develop other diseases because several risk factors, such as female sex, physical inactivity, obesity and lower socioeconomic status (SES), are common to many conditions.[4–6] Comorbidities increase the risk of hospitalisation, polypharmacy, premature death and substantially extend the burden on the healthcare system.[9] In patients with OA, the presence of comorbidities also has impacts on disease severity with increased pain, inferior physical functioning[10] and worsening of health-related quality of life,[11] which may affect the treatment outcome. An increased number of comorbidities has shown to be associated with both a greater deterioration of OA in form of function loss and pain severity,[10] but temporal relationship and causality are yet to be determined.[12]

First-line intervention for hip and knee OA comprises patient education and exercise combined with weight loss when necessary,[13–15] and is recommended regardless of the severity of the disease.[16 17] This treatment, when delivered in a real-world setting, has shown to alleviate OA symptoms, give patients a satisfactory quality of life, decrease willingness to undergo surgery, reduce use of OA medication and reduce sick leave.[18] However, those results include all individuals, regardless of the prevalence of comorbidities that may affect their outcome. To evaluate and improve treatment outcomes from first-line intervention for hip and knee OA, the magnitude of comorbidities should be described. The aim of the present study was to determine comorbidities, including distribution depending on age, of a population with hip and/or knee OA who were referred to first-line intervention in comparison with the general population. A secondary aim was to evaluate if differences in comorbidities were associated with socioeconomic inequalities.

## METHODS
### Study design and participants
This observational register-based study included all patients with hip and/or knee OA (n=72 069) in the Better Management of Patients with Osteoarthritis (BOA) national quality register between 2008 and 2016 (table 1). The BOA Register, register and evaluate patients in a standardised national first-line intervention programme for hip and knee OA according to international guidelines,[14 15] and provided by health professionals in primary care in Sweden.[17] Patients included in the present study were registered before participation (index date) and included regardless of compliance with treatment. They had sought help for hip and/or knee pain in primary healthcare and had been referred to the first-line intervention programme after a confirmed clinical or radiographic

**Table 1** Characteristics of the OA population from the BOA Register and the reference cohort and differences in comorbidities between the populations identified by the comorbidity indices

| | OA population (n=72 069) | Reference cohort (n=216 207) |
|---|---|---|
| Age (years) mean (SD) | 66.4 (9.6) | 66.4 (9.6) |
| Women, % (n) | 69 (49 494) | 69 (148 482) |
| **Most affect joint in the OA population, % (n)** | | |
| Hip | 32 (22 703) | |
| Knee | 68 (49 366) | |
| OA in multiple joints,* % (n) | 62 (44 384) | |
| *Missing* | *n=443* | |
| Born outside Sweden, % (n) | 9 (6474) | 13 (28 554) |
| *Missing* | *n=0* | *n=12* |
| Married, % (n) | 59 (42 359) | 54 (117 616) |
| *Missing* | *n=9* | *n=162* |
| Educational level, % (n) | | |
| Low (≤9 years) | 22.5 (16 276) | 28.5 (61 212) |
| Medium (10–14 years) | 60.5 (43 492) | 56 (119 416) |
| High (≥15 years) | 17 (12 111) | 15.5 (33 260) |
| *Missing* | *n=190* | *n=2319* |
| **Comorbidities indices** | | |
| *Missing* | *n=0* | *n=0* |
| RxRisk Index† median (IQR); min-max scores | 3 (4); 0–18 | 2 (4); 0–18 |
| 0 (no comorbidities), % (95% CI) | 16 (15.7 to 16.3) | 23 (22.8 to 23.2) |
| 1–3 | 44 (43.6 to 44.4) | 40.5 (40.3 to 40.7) |
| ≥4 | 40 (39.6 to 40.4) | 36.5 (36.3 to 36.7) |
| Elixhauser Comorbidity Index, median (IQR); min-max scores | 0 (1); 0–11 | 0 (1); 0–13 |
| 0 (no comorbidities), % (95% CI) | 69.5 (69.2 to 69.8) | 69 (68.8 to 69.2) |
| 1 | 16.5 (16.2 to 16.8) | 16 (15.8 to 16.2) |
| ≥2 | 14 (13.7 to 14.3) | 15 (14.8 to 15.2) |
| Charlson Comorbidity Index, median (IQR); min-max scores | 0 (0); 0-11 | 0 (0); 0-15 |
| 0 (no comorbidities), % (95% CI) | 81 (80.7 to 81.3) | 78 (77.8 to 78.2) |
| 1 | 9.5 (9.3 to 9.7) | 10 (9.9 to 10.1) |
| ≥2 | 9.5 (9.3 to 9.7) | 12 (11.9 to 12.1) |
| **Comorbidities (≥1) grouped by age, % (95% CI)†** | | |
| <45 years (OA n=1583; ref n=4749) | 65 (62.7 to 67.3) | 51 (49.6 to 52.4) |
| 46–65 years (OA n=29 225; ref n=87 675) | 79 (78.5 to 79.5) | 69 (68.7 to 69.3) |
| 66–80 years (OA n=36 993; ref n=110979) | 90 (89.7 to 90.3) | 84.5 (84.3 to 84.7) |
| >81 years (OA n=4268; ref n=12 804) | 96 (95.4 to 96.6) | 94 (93.6 to 94.4) |

*Osteoarthritis in multiple joints, identified as B or C Charley Class, a self-reported classification of musculoskeletal impairment. Class A indicates unilateral hip or knee OA, class B; bilateral hip or knee OA and class C; multiple joint OA or some other condition affecting the patient's ability to walk.
†The conditions 'inflammation/pain' and 'pain' are excluded from the score count.
BOA, Better Management of Patients with Osteoarthritis; n, numbers; OA, osteoarthritis.

diagnosis of OA following the Swedish National Board of Health and Welfare recommendations.[19–21] Patients with joint problems caused by another condition (eg, sequelae of hip fractures, chronic widespread pain, inflammatory joint diseases or cancer), patients who have undergone total joint replacement within the previous 12 months or other surgery of the knee or hip joint within the previous 3 months do not meet the criteria for inclusion and are therefore excluded from registration in the BOA Register.[17] The standardised first-line intervention for OA, the development of the BOA Register and the inclusion and exclusion criteria for registration of patients in the register have been described in detail previously.[17 22]

Without involvement from the researchers, Statistics Sweden (a government agency) randomly selected a reference cohort (n=216 207) of individuals from the general Swedish population for comparison using the Swedish Total Population Register (TPR)[23] (table 1). Those individuals had never been included in the BOA Register. To minimise confounding due to differences in access to care, the populations were matched for year of birth, sex and place of residence (counties in Sweden, n=21), at the index date in the BOA Register. Three reference individuals were identified for each patient in the OA population. The study followed the recommendations according to Strengthening the Reporting of Observational Studies in Epidemiology.[24]

### Data source
By using the 10-digit personal identity number (PIN), a unique number assigned to all Swedish residents at birth or immigration,[25] data on comorbidities from the index date and 3 years before that date, were obtained from two nationwide individual-based registers at the National Board of Health and Welfare: (1) the Swedish National Inpatient Register, which has an overall positive predictive value for diagnoses of 85%–95%; and (2) the Swedish Prescribed Drug Register, with data on expenditure of prescribed drugs in the entire Swedish population.[26 27] Socioeconomic individual-level data for the same year as the index date were obtained from the TPR and the Longitudinal Integration Database for Health Insurance and Labour Market Studies at Statistics Sweden.[23 28] Only the PIN and data on the most affected joint (hip or knee) were extracted from the BOA Register. Details of the different registers and the linkage of data between them, together with the creation and matching of a reference cohort from the general Swedish population, as well as a flowchart of the study participants from the BOA Register and their matched references have been described previously.[22 29]

### Identification of comorbidities
When describing the OA population and the reference cohort, we use the term comorbidities throughout the present study, defined as 'presence of an additional disease in relation to an index disease in one individual', even though a more correct definition regarding the

reference cohort would be morbidity or multimorbidities.[30] Comorbidities were identified using three different comorbidity indices: (1) the RxRisk Index,[31] (2) the Charlson Comorbidity Index[32] and (3) the Elixhauser Comorbidity Index,[33] and are presented as (1) individual diseases, (2) disease categories and (3) scores for each of the indices. Together, the three indices identified 66 individual diseases, which were grouped into 11 different disease categories (table 2).[34 35] When the scores for the indices were calculated, the unweighted versions of all three indices were used; for example, a score of 2 indicates that the patient had two comorbidities.[31 36 37]

The RxRisk Index uses data on prescriptions of drugs to identify the presence of 43 individual diseases (table 2) based on the Anatomical Therapeutic Chemical (ATC) classification system of the drugs, defined according to the WHO.[31] In this study, data from the Swedish Prescribed Drug Register were used, and the specific ATC codes for each of the 43 comorbidities are presented in online supplemental file 1. Scores for the index were categorised as 0, 1–3 or ≥4 comorbidities.

To calculate the Charlson Comorbidity Index[32] and the Elixhauser Comorbidity Index,[33] we used data on International Classification of Diseases, 10th revision (ICD-10) codes from the Swedish National Inpatient Register. The Charlson Comorbidity Index identifies 17 comorbidities and the Elixhauser Comorbidity Index identifies 30 comorbidities (table 2, online supplemental file 2). Scores for both indices were categorised as 0, 1 or ≥2 comorbidities.

### Indicators of socioeconomic status
Highest educational level achieved, country of birth and marital status were used as proxies for SES. Education was divided into three categories: low (≤primary school, 0–9 years), medium (secondary school plus up to <3 years postsecondary education, 10–14 years) and high (postsecondary education ≥3 years, ≥15 years). Country of birth was categorised as Sweden, the Nordic countries, Europe or others, and marital status as married (including registered partner) or not married.[29]

### Statistical analysis
Comorbidities in the OA population compared with the reference cohort are described using frequencies, proportions or medians and IQRs depending on the nature of the data. The OA population was also analysed separately grouped by age (<45, 46–65, 66–80 and >81 years) and by most affected joint (hip or knee) and compared with their respective matched references.

Two conditions that were identified by the RxRisk Index, inflammation/pain (treated with anti-inflammatory medication) and pain (treated with narcotics), were excluded in all calculations of the RxRisk Index, and from the analyses of the disease category, musculoskeletal and pain-related diseases, because they were considered as index diseases for this OA population.

**Table 2** Comorbidities of patients with osteoarthritis (OA) from the BOA Register, compared with the general population

| | RxRisk Index | | Elixhauser Comorbidity Index | | Charlson Comorbidity Index | |
|---|---|---|---|---|---|---|
| | OA | Ref | OA | Ref | OA | Ref |
| **Cancer** | 6% | 6% | | | | |
| Lymphoma | | | 0.5% | 0.5% | | |
| Malignancies | 0.5% | 0.5% | | | 5.5% | 6% |
| Metastatic cancer | | | 0.5% | 1% | 0.5% | 1% |
| Solid tumour without metastasis | | | 5% | 5.5% | | |
| **Cardiovascular/blood** | 59% | 54% | | | | |
| Anticoagulation agents/coagulopathy | 10% | 9% | 0.5% | 0.5% | | |
| Antiplatelet agents | 19% | 19% | | | | |
| Arrhythmias | 2% | 2% | 6% | 6% | | |
| Cerebrovascular disease | | | | | 2.5% | 3.5% |
| Congestive heart failure | 21.5% | 21% | 1.5% | 2.5% | 1.5% | 2.5% |
| Hyperlipidaemia | 28% | 26.5% | | | | |
| Hypertension | 31% | 27% | 15% | 14.5% | | |
| Ischaemic heart disease: angina | 7% | 6.5% | | | | |
| Ischaemic heart disease: hypertension | 34% | 32% | | | | |
| Myocardial infarction | | | | | 2.5% | 3% |
| Peripheral vascular disease | | | 1% | 1.5% | 1% | 1.5% |
| Pulmonary circulation disorders | | | 0.5% | 0.5% | | |
| Pulmonary hypertension | 0% | 0% | | | | |
| Valvular disease | | | 1.5% | 1.5% | | |
| **Endocrine** | 19% | 19% | | | | |
| Diabetes (uncomplicated) | 9% | 10% | 4.5% | 5% | 4.5% | 5% |
| Diabetes (complicated) | 1.5% | 1% | 1.5% | 2% | 1.5% | 2% |
| Hyperthyroidism | 0% | 0% | | | | |
| Hypothyroidism | 10.5% | 9.5% | 2% | 2% | | |
| Pancreatic insufficiency | 0.5% | 0.5% | | | | |
| **Gastrointestinal** | 35% | 28% | | | | |
| Gastro-oesophageal reflux disease | 32.5% | 25% | | | | |
| Irritable bowel syndrome | 1.5% | 1% | | | | |
| Liver disease (mild) | | | 0.5% | 0.5% | 0.5% | 0.5% |
| Liver disease (severe) or failure | 4% | 4.5% | | | 0% | 0% |

Continued

**Table 2** Continued

| | OA | Ref | RxRisk Index OA | RxRisk Index Ref | Elixhauser Comorbidity Index OA | Elixhauser Comorbidity Index Ref | Charlson Comorbidity Index OA | Charlson Comorbidity Index Ref |
|---|---|---|---|---|---|---|---|---|
| Peptic ulcer disease | | | | | 0.5% | 0.5% | 0.5% | 0.5% |
| **Musculoskeletal/pain related** | **70%** | **45%** | | | | | | |
| **- Excluding pain and inflammation/pain** | **11%** | **10%** | | | | | | |
| Gout | | | 2.5% | 2.5% | | | | |
| Migraine | | | 2.5% | 2% | | | | |
| Osteoporosis/Paget's | | | 4.5% | 4.5% | | | | |
| *Pain* | | | 31.5% | 23% | | | | |
| *Inflammation/pain* | | | 58.5% | 31.5% | | | | |
| Rheumatic disease | | | | | | | 1.5% | 2% |
| Rheumatoid arthritis/collage vascular diseases | | | | | 2% | 2.5% | | |
| **Neurological** | **7%** | **8%** | | | | | | |
| Dementia | | | 0% | 1% | 0% | 0% | 0% | 1% |
| Epilepsy | | | 4% | 4.5% | | | | |
| Paralysis | | | | | 0% | 0.5% | | |
| Hemiplegia or paraplegia | | | | | | | 0% | 0.5% |
| Parkinson's disease | | | 2.5% | 2% | | | | |
| Other neurological disorders | | | | | 1% | 2% | | |
| **Nutritional/obesity** | **3%** | **3%** | | | | | | |
| Blood loss anaemia | | | | | 0% | 0% | | |
| Deficiency anaemia | | | | | 0.5% | 0.5% | | |
| Fluid and electrolyte disorders | | | | | 0.5% | 1% | | |
| Hyperkalaemia | | | 0% | 0% | | | | |
| Malnutrition | | | 0% | 0% | | | | |
| Obesity | | | | | 2% | 1% | | |
| Weight loss | | | | | 0% | 0% | | |
| **Psychological/behavioural** | **23%** | **24%** | | | | | | |
| Alcohol abuse/dependency | | | 0.5% | 0.5% | 0.5% | 1% | | |
| Anxiety | | | 8.5% | 10% | | | | |
| Bipolar disorder | | | 0.5% | 0.5% | | | | |
| Depression | | | 17.5% | 17.5% | 2% | 2.5% | | |
| Drug abuse | | | | | 0% | 0.5% | | |

Continued

**Table 2** Continued

| | OA | Ref | RxRisk Index | | Elixhauser Comorbidity Index | | Charlson Comorbidity Index | |
|---|---|---|---|---|---|---|---|---|
| | | | OA | Ref | OA | Ref | OA | Ref |
| Psychotic illness/psychoses | | | 0.5% | 1.5% | 0% | 0.5% | | |
| Smoking cessation | | | 1.5% | 1.5% | | | | |
| **Renal/urological** | **9%** | **7%** | | | | | | |
| Benign prostatic hyperplasia | | | 4.5% | 3.5% | | | | |
| Incontinence | | | 4.5% | 3% | | | | |
| Renal disease/failure | | | 0.5% | 0.5% | 0.5% | 1% | 0.5% | 1% |
| **Respiratory** | **12%** | **11%** | | | | | | |
| Chronic pulmonary disease | | | | | 3% | 3.5% | 3% | 3.5% |
| Chronic airways disease | | | 11% | 9.5% | | | | |
| **Miscellaneous** | **37%** | **31%** | | | | | | |
| Allergies | | | 24% | 19.5% | | | | |
| Glaucoma | | | 4% | 4% | | | | |
| HIV/AIDS | | | 0% | 0% | 0% | 0% | 0% | 0% |
| Psoriasis | | | 1.5% | 1.5% | | | | |
| Steroid-responsive conditions | | | 16.5% | 13.5% | | | | |
| Transplant | | | 0% | 0% | | | | |

Comorbidities presented as disease categories and as individual comorbidities, identified by the RxRisk Index, the Elixhauser Comorbidity Index and the Charlson Comorbidity Index, for the OA population (n=72 069) compared with their references from the general population (n=216207). All numbers are reported as percentages (%). The numbers marked in bold are the percentage of each population that has been identified with at least one of the individual diseases included in that disease category. The other numbers are the percentage of each population that has been identified with each individual disease by each of the indices.

BOA, Better Management of Patients with Osteoarthritis.

Logistic regression models were used to evaluate the eventual effect of SES on the prevalence of comorbidities. Both crude and adjusted analyses were performed for socioeconomic indicators (educational level, country of birth and marital status), calculating the OR with the 95% CI. For those who were included in the BOA Register up to 2015 and their matched references, we also had access to data on other socioeconomic indicators (income, employment and family type). Secondary analyses were therefore performed to evaluate if those indicators affected the results, but because they did not, we chose only to use those indicators for SES that we had access to for the total study population (presented in online supplemental file 3).

Nearly two-thirds of the OA population but only one-third of their references had prescriptions for anti-inflammatory medications (ATC codes M01A–M01H), drugs that increase the risk of developing gastrointestinal comorbidities,[38] and thereby often are prescribed preventively in combination with gastroprotective drugs (ATC codes A02BA01–A02BX05). Therefore, we also chose to do separate analyses of those who were not using anti-inflammatory drugs to try to establish the real prevalence of gastro-oesophageal diseases.

The merging of data from the National Board of Health and Welfare and the creation of the database were performed using SAS V.9.4 TS Level 1MS. All statistical analyses were performed with IBM SPSS Statistics for Windows, V.25.0 (IBM Statistics).

## Patient and public involvement

Patients were not involved in the development of the research question or the design of this study.

The study was registered at ClinicalTrials.gov. The study followed the legal and ethical frameworks of informed consent in register-based research, as described by Swedish law and ethical boards.[39] All patients in the BOA Register received oral and written information about their registration, including information that data may be used in research. Regarding the reference cohort, informed consent was waived according to standard practice in Sweden, since no individuals in this population-based cohort could be identified. Statistics Sweden replaces each PIN with a serial number to anonymise data before returning it to the researchers.

## RESULTS
### Study participants

The characteristics of the participants in the study are described in table 1. The mean age was 66.4 years (SD, 9.6 years) and 69% were women. In the OA population, 32% were registered with OA most affecting the hip joint and 68% most affecting the knee joint.

### Prevalence of comorbidities

A higher proportion of individuals in the OA population (85%) had comorbidities than in the reference cohort; 78% of the references were identified with at least one comorbidity. Of the 66 comorbidities that could be traced, 23 had a low prevalence with ≤1% in each of the populations (table 2). Cardiovascular/blood diseases were the most common comorbidities in the disease category, with 59% of this OA population and 54% of their references identified with at least one of those, followed by gastrointestinal diseases (35% of OA and 28% of the references).

In the OA population, 32.5% were identified (by the RxRisk Index) as having medication prescribed for gastro-oesophageal reflux disease compared with 25% of the reference cohort. Separate analyses excluding those without concomitant treatments with anti-inflammatory medications showed that the prevalence of gastro-oesophageal reflux disease evened out between the two populations, but was still higher in the OA population (24%) than among the references (19%).

A fourth of both populations had psychological/behavioural comorbidities; depression (17.5% in both populations) and anxiety (8.5% in the OA population and 10% in the reference cohort) were the most common with a similar representation in both populations (table 2). Also the prevalence of endocrine diseases such as diabetes was equally distributed in the two populations (19%).

When scores were calculated for each of the three indices, the OA population was identified with more comorbidities by the RxRisk Index, compared with their references; with the Elixhauser Comorbidity Index, the prevalence of comorbidities was similar between the two populations; and with the Charlson Comorbidity Index, the OA population showed a slightly lower prevalence of comorbidities compared with the reference cohort (table 1).

### Associations between the populations and comorbidities

The OA population had a higher risk of having at least one other comorbidity (OR, 1.62; 95% CI 1.59 to 1.66), but the odds then decreased for additional comorbidities, with an OR of 1.36 (95% CI 1.34 to 1.39) for having ≥2 and 1.23 (95% CI 1.21 to 1.26) for having ≥3 comorbidities. Adjustment for SES led to a slightly higher OR overall than the crude value (table 3).

Being part of this OA population was also associated with a higher odds of having comorbidities in several of the disease categories (cardiovascular/blood, endocrine, gastrointestinal, renal/urological, respiratory and miscellaneous), but also statistically significant lower ORs for cancer, neurological, nutritional/obesity and psychological/behavioural diseases. The results when adjusted for SES were similar to the crude ORs (figure 1 and table 3).

### Separate analyses of subgroups

Grouping the populations by age, the OA population had a higher presence of comorbidities (≥1) in the younger age groups compared with their matched references, whereas in the oldest age group, the prevalence were similar in the two populations (table 1). The OA population also

**Table 3** OR of having comorbidities for the OA population in the BOA Register compared with the reference cohort

| | Total study population | | Hip OA | | Knee OA | |
|---|---|---|---|---|---|---|
| | Crude OR (95% CI) | Adjusted OR (95% CI)* | Crude OR (95% CI) | Adjusted OR (95% CI)* | Crude (95% CI) | Adjusted OR (95% CI)* |
| **Individual comorbidities†** | | | | | | |
| ≥1 | 1.62 (1.59 to 1.66)§ | 1.67 (1.63 to 1.71)§ | 1.48 (1.42 to 1.54)§ | 1.53 (1.46 to 1.59)§ | 1.69 (1.65 to 1.74)§ | 1.73 (1.69 to 1.78)§ |
| ≥2 | 1.36 (1.34 to 1.39)§ | 1.41 (1.38 to 1.44)§ | 1.26 (1.22 to 1.30)§ | 1.31 (1.27 to 1.35)§ | 1.41 (1.38 to 1.45)§ | 1.46 (1.43 to 1.49)§ |
| ≥3 | 1.23 (1.21 to 1.26)§ | 1.28 (1.26 to 1.31)§ | 1.14 (1.11 to 1.17)§ | 1.19 (1.16 to 1.23)§ | 1.28 (1.25 to 1.31)§ | 1.33 (1.30 to 1.36)§ |
| **Disease categories** | | | | | | |
| Cancer | 0.89 (0.86 to 0.92)§ | 0.89 (0.85 to 0.92)§ | 0.90 (0.85 to 0.96)§ | 0.89 (0.84 to 0.95)§ | 0.88 (0.84 to 0.92)§ | 0.88 (0.84 to 0.92)§ |
| Cardiovascular/blood | 1.22 (1.20 to 1.24)§ | 1.26 (1.24 to 1.28)§ | 1.15 (1.11 to 1.18)§ | 1.19 (1.15 to 1.23)§ | 1.26 (1.23 to 1.29)§ | 1.29 (1.27 to 1.32)§ |
| Endocrine | 1.04 (1.02 to 1.06)§ | 1.09 (1.06 to 1.11)§ | 0.97 (0.93 to 1.01) | 1.02 (0.98 to 1.06) | 1.08 (1.05 to 1.10)§ | 1.12 (1.09 to 1.15)§ |
| Gastrointestinal | 1.38 (1.35 to 1.40)§ | 1.43 (1.41 to 1.46)§ | 1.31 (1.27 to 1.35)§ | 1.37 (1.33 to 1.42)§ | 1.41 (1.38 to 1.44)§ | 1.46 (1.43 to 1.50)§ |
| Musculoskeletal/pain-related‡ | 1.03 (1.00 to 1.06) | 1.04 (1.01 to 1.07)§ | 1.02 (0.97 to 1.07) | 1.02 (0.97 to 1.08) | 1.04 (1.00 to 1.07) | 1.04 (1.01 to 1.08) |
| Neurological | 0.79 (0.76 to 0.82)§ | 0.82 (0.79 to 0.84)§ | 0.74 (0.69 to 0.78)§ | 0.76 (0.71 to 0.80)§ | 0.82 (0.78 to 0.85)§ | 0.84 (0.81 to 0.88)§ |
| Nutritional/obesity | 0.92 (0.88 to 0.97)§ | 0.97 (0.92 to 1.02) | 0.75 (0.68 to 0.83)§ | 0.79 (0.72 to 0.87)§ | 1.01 (0.95 to 1.07) | 1.06 (0.99 to 1.12) |
| Psychological/behavioural | 0.96 (0.94 to 0.98)§ | 0.99 (0.97 to 1.01) | 0.94 (0.90 to 0.97)§ | 0.97 (0.93 to 1.00) | 0.96 (0.94 to 0.99)§ | 1.00 (0.97 to 1.02) |
| Renal/urological | 1.30 (1.26 to 1.34)§ | 1.32 (1.28 to 1.36)§ | 1.24 (1.18 to 1.31)§ | 1.27 (1.20 to 1.34)§ | 1.33 (1.29 to 1.38)§ | 1.34 (1.30 to 1.40)§ |
| Respiratory | 1.16 (1.13 to 1.19)§ | 1.18 (1.14 to 1.21)§ | 1.11 (1.06 to 1.16)§ | 1.13 (1.08 to 1.18)§ | 1.18 (1.15 to 1.22)§ | 1.20 (1.16 to 1.24)§ |
| Miscellaneous | 1.29 (1.27 to 1.31)§ | 1.29 (1.27 to 1.31)§ | 1.23 (1.19 to 1.27)§ | 1.22 (1.19 to 1.26)§ | 1.32 (1.29 to 1.35)§ | 1.32 (1.29 to 1.35)§ |

*Adjusted for socioeconomic status (educational level, country of birth and marital status).
†Individual comorbidities identified by summing the RxRisk Index, the Elixhauser Comorbidity Index and the Charlson Comorbidity Index.
‡The conditions inflammation/pain and pain identified by the RxRisk Index were excluded from the analyses.
§Statistically significant results.
BOA, Better Management of Patients with Osteoarthritis; OA, osteoarthritis.;

showed an overall higher odds of having one or more comorbidities for all age groups, the younger, the higher ORs. There were no differences in OR for ≥1, ≥2 or ≥3 comorbidities in the youngest age group (aged <45 years). In contrast, the ORs for ≥2 or ≥3 comorbidities were continuously lower than having only ≥1 comorbidity in the older age groups (aged 46–>81 years) (figure 2).

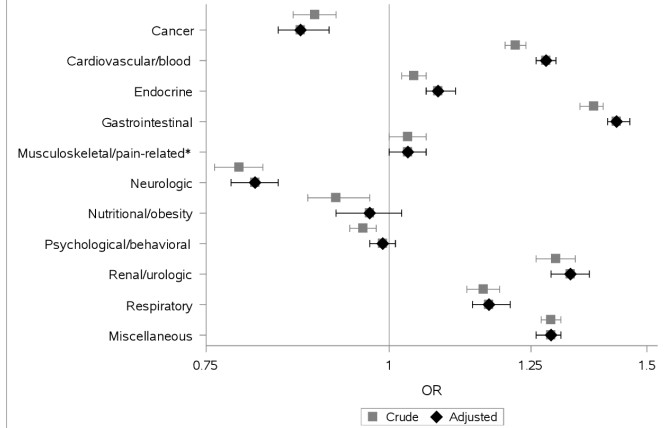

**Figure 1** Forest plot with OR and 95% CI for having comorbidities for the osteoarthritis population compared with the reference cohort, grouped in disease categories. Reported in crude OR, and in OR adjusted for socioeconomic status (educational level, country of birth and marital status). *The conditions inflammation/pain and pain, identified by the RxRisk Index, were excluded from the analyses.

When grouping the OA population by most affected joint (hip/knee) and then comparing them with their respective matched references, those registered with knee OA had more comorbidities than those registered with hip OA (table 3). Also in the separate analyses of the populations in subgroups, adjustment for SES had only a minor effect on the estimates (figure 2 and table 3).

## DISCUSSION

This study shows that individuals referred to a standardised national first-line intervention for OA in Sweden had an overall higher prevalence of comorbidities than their matched references from the general population. The study also shows that there was a higher prevalence of comorbidities among younger individuals in the OA population compared with their references, whereas the prevalence was similar for older individuals, which could indicate that it is probably important to initiate first-line intervention early to reduce the risk of developing related disease.[17] Patients in the OA population registered with knee OA had more comorbidities than those registered with hip OA. This may be partly explained by the fact that obesity is more common among those registered with knee OA compared with hip OA in the BOA Register,[18] which is also a risk factor for the development of several other diseases.[5 6]

Previously, we have shown an overall higher SES in this OA population from the BOA Register compared

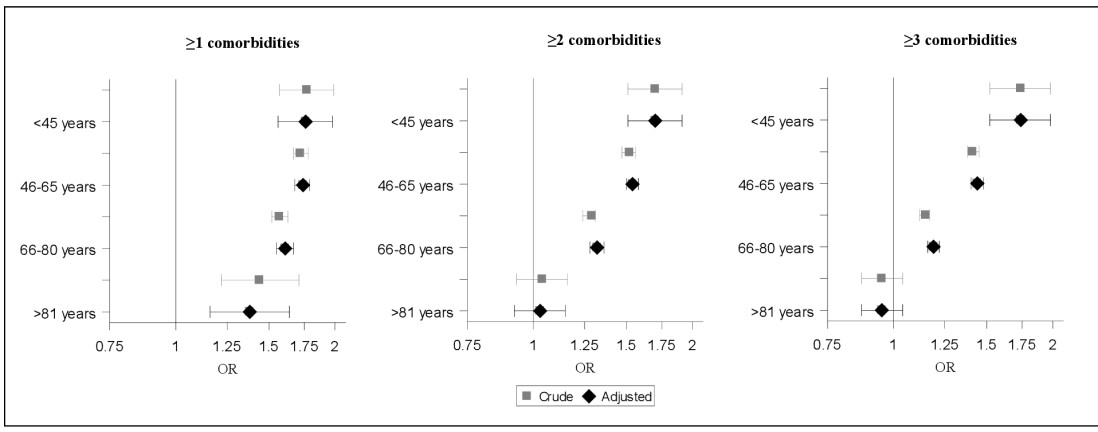

**Figure 2** Forest plot with OR and 95% CI for having ≥1 comorbidities for the osteoarthritis population compared with the reference cohort, grouped by age. Reported in crude OR, and in OR adjusted for socioeconomic status (educational level, country of birth and marital status).

with that of the general Swedish population.[29] From the results in the present study, we can now report that the inequalities in SES were not associated with differences in comorbidities, but must be related to other factors. It is well known that there is a connection between SES and several common chronic diseases, with poorer health among more socioeconomically disadvantaged individuals,[40 41] and that OA is in general more common among individuals with lower SES.[42 43] The reasons why this OA population had a higher SES, but had more comorbidities than the general population, are probably highly dependent on the association that exists between OA and other diseases.[4] In addition, it could also be the effect of individuals' lifestyle and healthcare seeking behaviours; individuals with lower SES tends to seek healthcare less often, which could result in both late diagnosis and delayed treatment.[44] It has also been shown previously that more disadvantaged individuals with OA have poorer access to both non-surgical healthcare services, such as self-management education, physiotherapy and medication and to joint replacement surgery.[45 46] In the present study, the OA population showed a higher prevalence of comorbidities compared with the reference cohort, and most comorbidities for both populations were detected from data on prescriptions of drugs and by the RxRisk Index, whereas data from ICD-10 codes for inpatient care showed slightly more comorbidities among the reference cohort. It could be argued that the BOA population probably to a higher extent includes individuals who take responsibility for their health, who seek help for their OA problems and at the right level of care, and who take the medications that are prescribed for them, and that they are thus also more represented and appear in registers that include prescription of drugs.

The differences in comorbidities between the populations in the present study were not of the same magnitude as in a recent review by Swain *et al*.[4] They reported that 67% of individuals with OA were identified with ≥1 other chronic condition, which was 20% higher than among non-OA individuals. Pihl *et al*[47] identified self-reported

comorbidities among approximately 50% of a similar Danish OA population, before participation in a first-line intervention programme for hip and knee OA, and that the presence of comorbidities were not associated with outcomes after the intervention. However, differences in prevalence of comorbidities between studies may be due to several factors, making comparison challenging, such as different study designs, the sample characteristics, methods of identifying comorbidities and the number of comorbidities analysed. Knowledge of how OA and comorbidities associate is still largely lacking and further research is needed.[4] Instead of comparing the overall prevalence of comorbidities in this OA population with results from other studies, it may be more interesting to compare prevalence divided into different groups of diseases. In the present study, we identified that cardiovascular/blood diseases was the most common disease category in both populations, with a higher prevalence in the OA population compared with the reference cohort. These results are supported in a previous review by Hall *et al*,[48] who established that individuals with OA had a significantly higher prevalence of cardiovascular diseases, especially heart failure and ischaemic heart disease, compared with matched controls. The prevalence of diabetes[49] and depression[7] has also been shown to be higher in OA populations compared with general populations. Those results were not supported by this study; we instead identified similar prevalence in both populations. Our separate analyses excluding individuals with an anti-inflammatory prescription showed that the difference in the prevalence of gastro-oesophageal reflux disease was reduced but remained higher in this OA population. This may be due to greater use of over-the-counter anti-inflammatory medication, which can be assumed to be higher among people with OA. The odds of having cancer or neurological diseases was lower in the OA population than the reference cohort, but those results are probably the effect of the criteria for registration in the BOA Register, which excludes individuals with OA who also have other severe diseases.[17]

In the present study, we used three comorbidity indices to make it possible to identify several diseases, and from different data sources. The Charlson Comorbidity Index and the Elixhauser Comorbidity Index were developed to capture comorbidities in inpatient care using diagnostic ICD-10 codes,[36] and are commonly used in orthopaedics research,[34] whereas the RxRisk Index identifies comorbidities from prescription of drugs by ATC codes and can be used in predominately outpatient settings,[31] such as this specific OA population. All three indices have been used previously to study comorbidities in OA populations.[34 35] The possibility to use comorbidity indices and summarise them into scores is necessary when handling large administrative data sets, but there is no gold standard with regard to which index should be used primarily. Which method that is best suited to describe comorbidities depends on the circumstances, and some comorbidities are better detected using medication-based data and others with diagnosis-based data.[34 35] For example; the diagnose diabetes, can by the Charlson Comorbidity Index and the Elixhauser Comorbidity Index also include individuals who are treated without diabetes medication, which is not the case with the RxRisk Index. With the Charlson Comorbidity Index, complications in patients with diabetes should be chronical, but not necessarily so according to Elixhauser Comorbidity Index.[36] Therefore, even if the same comorbidities are identified by different indices, they should not be compared between the different indices, only between different populations. In the present study with an aim to describe the magnitude of different comorbidities, we chose to combine the three indices to capture the overall burden of the comorbidities, instead of only focusing on separate scores for each index.

### Limitations and strengths

There are some limitations in the present study that should be highlighted. To seek healthcare for a condition like OA may increase the probability to also be registered for other diagnoses, and/or receiving prescriptions of drugs for other conditions. Obesity increases the risk of developing several chronic conditions including OA[50] and may be a confounding factor of the results. Since we only had access to data on body mass index for the OA population, but not for the reference cohort, we could not adjust for obesity. We were only able to capture ICD-10 codes to calculate the Charlson Comorbidity Index and the Elixhauser Comorbidity Index from healthcare delivered through inpatient care and specialised outpatient visits. Data on primary healthcare and information on healthcare provided by professionals other than physicians are not available on a national level in Sweden. However, both the Charlson Comorbidity Index and the Elixhauser Comorbidity Index were developed primarily to capture comorbidities in inpatient care.[36]

Another limitation to bear in mind when interpreting the results from the present study is that the criteria for registration in the BOA Register exclude individuals who have other severe diseases.[17] This may contribute to underestimating comorbidities, because we can assume that this OA population probably has less comorbidities than general OA populations who are at the same stage of their disease, when seeking healthcare and structured OA treatments, such as the OA population in the present study. The contradictive results that the Charlson Comorbidity Index displayed a slightly higher prevalence of comorbidities in the reference cohort could also be an effect of this exclusion of individuals with severer diseases in the BOA Register, since the diseases detected by Charlson consistently are of a somewhat more serious nature. In addition, there could be individuals with OA in the reference cohort, which may affect the results; OA is common in those age groups, it is difficult to define the onset of the disease,[51] many individuals with OA symptoms do not seek healthcare[52] or are not referred to first-line intervention for OA. For example, it has been estimated that during 2014, the BOA Register reached 17% of all individuals who sought outpatient care with the diagnosis of hip or knee OA.[53]

Despite these circumstances, we could identify a higher prevalence of several diseases in the OA population compared with the age-matched, sex-matched and residence-matched reference cohort. The results from the present study are applicable on OA individuals referred to first-line intervention, but are also in line with several previous studies,[4 5 48] describing comorbidities among individuals with OA compared with general populations.

Even though more recent guidelines on OA treatment include recommendations for individuals with comorbidities,[15] they are often excluded from exercise treatment,[54] with the risk of being left without any treatment. However, OA is not a single-joint disease and to improve treatment outcomes from first-line intervention for hip and knee OA, it is important to not only focus on specific OA treatments but also on increased overall health.

### CONCLUSION

Comorbidities were more common among individuals referred to first-line intervention for hip and knee OA than among a matched reference cohort from the general population, especially among younger individuals and those with OA most affecting their knees. Based on the results from this study, socioeconomic status does not seem to affect the differences in comorbidities detected between this OA population and the reference cohort.

**Author affiliations**
[1]Unit of Physiotherapy, Department of Health, Medicine and Caring Sciences, Linköping University, Linköping, Sweden
[2]Department of Physiotherapy, Rehabilitation Centre, Ryhov County Hospital, Jönköping, Sweden
[3]Department of Medical and Health Sciences, Linköping University and Futurum - Academy for Health and Care, Region of Jönköping County, Jönköping, Sweden
[4]Department of Clinical Sciences Lund, Orthopaedics, Faculty of Medicine, Lund University, Lund, Sweden
[5]Centre of Registries, Västra Götaland, Gothenburg, Sweden

[6]Department of Orthopaedicst, Sahlgrenska Academy, University of Gothenburg, Gothenburg, Sweden

**Acknowledgements** The authors would like to acknowledge all participating patients, physiotherapists reporting data to the BOA Register and others involved in BOA.

**Contributors** KG, JK, ME and OR designed the study. KG and OR contributed to acquisition of data. CZ was responsible for merging of data from the BOA Register with data from the National Board of Health and Welfare and Statistics Sweden and for the creation of the research database. KG had full access to all data that were analysed in the study and takes, together with CZ, responsibility for the accuracy of the data analysis. All authors (KG, JK, ME, ADI, CZ, LD and OR) contributed in interpretation of data. KG wrote the first draft of the paper, which was critically revised by the other authors. All authors interpreted the findings and approved the final version of the manuscript before submission.

**Funding** The study was financially supported by AFA Insurance, Sweden, Futurum - Academy for Health and Care, Region Jönköping County, Sweden and the Medical Research Council of Southeast Sweden.

**Competing interests** LD is the co-founder and Chief Medical Officer of Joint Academy, a company that provides digital first-line intervention for patients with hip and knee osteoarthritis. The other authors declare that they have no conflicts of interest.

**Patient consent for publication** Not required.

**Ethics approval** The study was approved by the Regional Ethical Review Board in Gothenburg, Sweden (entry number 1059-16).

**Provenance and peer review** Not commissioned; externally peer reviewed.

**Data availability statement** Data may be obtained from a third party and are not publicly available. The dataset supporting the conclusions of this article is governed by Västra Götalandsregionen. The authors are not allowed to share the data. The data are available from Registercentrum, Västra Götalandsregionen (contact; boa@registercentrum.se) for researchers who meet the criteria for access to confidential data according to Swedish law.

**ORCID iDs**
Kristin Gustafsson http://orcid.org/0000-0003-3403-229X
Joanna Kvist http://orcid.org/0000-0003-3527-5488
Marit Eriksson http://orcid.org/0000-0002-1766-5899
Andrea Dell'Isola http://orcid.org/0000-0002-0319-458X
Leif E. Dahlberg http://orcid.org/0000-0001-7838-9464
Ola Rolfson http://orcid.org/0000-0001-6534-1242

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
