## [Reviewer comments · BMJ Open]

ARTICLE DETAILS

TITLE (PROVISIONAL)	Health status of individuals referred to first-line intervention for hip and knee osteoarthritis compared with the general population: an observational register-based study
AUTHORS	Gustafsson , Kristin; Kvist, Joanna; Eriksson, Marit; Dell'Isola, Andrea; Zhou, Caddie; Dahlberg, L; Rolfson, O

VERSION 1 – REVIEW

REVIEWER	Jamsen, Esa Hospital for Joint Replacement
REVIEW RETURNED	15-Mar-2021

GENERAL COMMENTS	Although the authors have obviously taken a great effort in combining and analyzing their data (according to previously published protocol - thank you for attaching it!) and in general reporting is fluent, there are severe issues related to the methodology, generalizability of the results and interpretation and discussing the results. Main concerns: 1. The study gives very limited - if any - new information. There are numerous studies about comorbidity in patients with osteoarthritis and the authors themselves cite several systematic reviews and meta-analyses and there is already one comparable study (Pihl et al.). The study mostly reports what was already known. The authors focus is on patients who receive "first-line intervention" who represent a subsample of all patients with OA and I do not see why previous studies would not apply to this patient population, too. I agree that in order to improve first-line intervention, it would be essential to know how comorbidity affects its results. The current study does not answer that question and I am looking forward for seeing the results of the outcome analysis.2. The generalizability of the results is unclear and obviously compromised. The study is nationwide but - as the authors indicate - includes only a part of all patients presenting with OA complaints ("during 2014, the BOA Register reached 17% of all individuals who sought outpatient care with the diagnosis of hip or knee OA"). Moreover, the protocol indicates that in 2014, only 69% of patients receiving the studies intervention have been registered. There were also exclusions for "severe comorbidities" like cancer so the study does not give population-based data (such major exclusion criteria should be mentioned in the methods section, reference to previous publications does not suffice). The
--

reference population may include persons with OA (but not registered to the studied BOA register) as well as patients with joint replacements (at least to my understanding). It is unclear if the BOA population had OA in multiple joints as only the dominating one has been registered.

3. The way how authors have dealt with comorbidity data is very problematic. It is great that they have used drug-data to supplement the often limited hospitalization data and the study also covers a large number of different conditions and is not limited to comorbidity indices. However, having OA and receiving care for it increases the odds for receiving care to other conditions as well and the comorbid conditions being registered. Table 2 shows that there are major differences in prevalences of different conditions. Hence, calculating prevalences of different types of comorbidities (CV disease, endocrine disease and so on) based on that any of registers have captured the conditions feels like counting apples with oranges. The results are currently dominated by the drug data (RxRisk) and there the problem is that the list of conditions includes many that could not be considered comorbidities (smoking cessation, pain, inflammation/pain). It is confusing that "with the Charlson Comorbidity Index, the OA population showed a slightly lower prevalence of comorbidities compared with the reference cohort" which opposes other results and the authors' conclusion. These limitations and their consequences have not been adequately in the discussion.

4. Some of the results are obvious consequences of inclusion criteria (like lower cancer in the BOA cohort) limiting the value of some of the findings. It is possible that some results that conflict with previous studies (depression, anxiety) are related to the methodology as well.

5. Obesity is a major confounding and/or explanatory factor in this study. However, it has been ignored and has not been discussed as limitation. To my understanding, BOA register includes BMI data but it was not used.

6. Even with the use of drug data, the prevalences of certain comorbidities is lower than one would expect (for example, diabetes). Partly this - and especially the lower prevalences with Elixhauser and Charlson indices - is likely related to the fact that comorbidities were registered only from three years before the index date. This selection is not reasoned in any way.

7. It is unclear how did the patients consent to participate. It feels unlikely that all would be willing to be registered to a quality register.

Minor concerns (include but are not limited to the following):

- Abstract could include more data about differences in individual comorbidities as well as about the age effect.
- Some of the references feel a bit old.
- Use of references is not fully adequate. For example, there are two studies about future projections of OA epidemiology are used

	as references for text related to symptoms of OA in the introduction.  - Differences in the results of the three different comorbidity tools should be discussed. - It is surprising that there are differences in baseline data like being born in Sweden and marital status although the selection criteria of reference population were not strict.
--	---

REVIEWER	Gronhaug, Gudmund østfold hospital trust, Department of physical medicine and rehabilitation
REVIEW RETURNED	17-Mar-2021

GENERAL COMMENTS	Thank you for inviting me to review this manuscript. I found it to be an interesting study to review and over all this is a well organised and well written paper. I have a few comments to the authors that, in my opinion, needs to be addressed. I will present my concerns in the line of order as they appear in the manuscript. Introduction Page 6 Line 45 - 58 is presenting the benefits of first line interventions for OA patients. As this is not the aim of the study I am not sure if this is needed in the introduction. I would like the authors to present more details on how and why OA patients are facing a possible downward spiral of comorbidities and loss of function and quality of life as a result of OA. Page 15 Indicators of socioeconomic status Using education as a socioeconomic marker is fine. Still, when working with statistics of Sweden I suppose it also was possible to get data on status of employment for the whole study group and not just the As OA patients are more often on sick leave and face threats of losing their job it would strengthen the study to use employment status as well as educational level as markers of SES. Page 16 line 1-8 Statistical analysis of other SES markers If employment did not affect the results different than the educational level it would be interesting to see the analysis and have them presented. Maybe as a sub group analyses for the group that is included before 2015? Line 13 - 27 Parts of this section belongs in the methods section and not the statistics section. Please check it. Line 43-44 This information is not about this study is it? It is OK not to involve patients in all studies. Page 20 separate analyses of subgroups
--

	It would be interesting to see a full sub-group analysis of comorbidities in the group of this who have affected BOTH hip and knee compared with only knee or hip and their matched controls. Discussion Although a lot of the discussion is based on assumptions I find it convincing and well balanced based on the findings. Still, in the limitations, or the discussion, it should be mentioned that it is likely that some of the controls probably also are OA patients. In my opinion this is likely to contribute to the results and should be mentioned. Another more philosophical question that might not be appropriate to discuss in this paper is how to define comorbidities in the control group. For the OA population one extra disease is a comorbidity. For the controls the first disease is not a comorbidity as it is not linked to a baseline diagnosis. This is to some extent taken into account by using the comorbidity indexes as I understand it. But it might be worth mentioning how the authors are overcoming this philosophical obstacle of counting diagnoses.
--	--

VERSION 1 – AUTHOR RESPONSE

Reviewer: 1

Dr. Esa Jansen, Hospital for Joint Replacement

Reply:

Thank you for your reflections on our study and your thoughts to further improve the value of the manuscript. Below we answer, comment by comment.

Main concerns:

1. The study gives very limited - if any - new information. There are numerous studies about comorbidity in patients with osteoarthritis and the authors themselves cite several systematic reviews and meta-analyses and there is already one comparable study (Pihl et al.). The study mostly reports what was already known. The authors focus is on patients who receive "first-line intervention" who represent a subsample of all patients with OA and I do not see why previous studies would not apply to this patient population, too. I agree that in order to improve first-line intervention, it would be essential to know how comorbidity affects its results. The current study does not answer that question and I am looking forward for seeing the results of the outcome analysis.

Reply:

First, we would like to start by thanking you for your valuable comments. We appreciate your input and for giving us the opportunity to clarify unclear aspects.

The BOA Register today in 2021, includes data from over 140 000 patients with knee and hip OA who are referred to a first-line OA intervention. To our knowledge, this is probably the world's largest collection of data on an OA population. In Sweden, the personal identity number (PIN) enable linking data from the BOA Register with data from other Swedish health and socioeconomic data sources. This provides an opportunity to study a range of factors that may influence the progression of OA and factors that can predict long-term outcomes of OA in the BOA population. However, the study design has limitations that we now hope are more clearly addressed in the discussion. Ofcourse, we would like to address all the proposed reflections and suggestions. Due to both limitations in data and scope of the present study, this is not possible, but we hope to report outcome analyses of comorbidities in future studies.

We have further chosen to reply to both comment 1 and 2 together since we believe these are interlinked. Please see our further reply after comment 2.

2. The generalizability of the results is unclear and obviously compromised. The study is nationwide but - as the authors indicate - includes only a part of all patients presenting with OA complaints ("during 2014, the BOA Register reached 17% of all individuals who sought outpatient care with the diagnosis of hip or knee OA"). Moreover, the protocol indicates that in 2014, only 69% of patients receiving the studies intervention have been registered. There were also exclusions for "severe comorbidities" like cancer so the study does not give population-based data (such major exclusion criteria should be mentioned in the methods section, reference to previous publications does not suffice). The reference population may include persons with OA (but not registered to the studied BOA register) as well as patients with joint replacements (at least to my understanding). It is unclear if the BOA population had OA in multiple joints as only the dominating one has been registered.

Reply:

We agree that the association between comorbidities and osteoarthritis (OA), has been studied previously but the focus has often been on single diseases such as cardio-vascular diseases or diabetes,[1, 2] or prevalence and patterns of multimorbidities in general primary care populations [3, 4] with OA as one of the included conditions. Few studies have, like the present study, focused on a wide range of different comorbidities at the same time. During 2020, Swain et al [5] published a systematic review of the literature on comorbidities in people with OA, compared with non-OA people. They reflected that; "Comorbidity research in OA is still at a preliminary stage and the evidence is yet to be accumulated" and that "the association between OA and other chronic conditions remains largely unknown".

Our population does represent a subsample of patients with OA, but OA is to a large extent a very heterogeneous disease and to be able to draw correct conclusions from outcome studies, the studied population needs to be specified, to also be able to determine if outcome results are generalizable. In our opinion, previous studies do not apply to this specific OA population, since it from comorbidity research performed so far, still is difficult to compare comorbidities between studies. As we have written in the discussion section, this is both due to differences in study design, characteristics of the included samples, and because there is no consensus in how comorbidities are identified and measured. Pihl et al,[6] is an example of a comparable subsample of OA patients (a study sample in a first-line intervention), but the comparison is still difficult due to differences in both patient characteristics (younger and more highly educated) and how comorbidities are identified (11 different self-reported conditions).

If the result from this present study can be generalized is an important reflection and we thank you for giving us the opportunity to respond and clarify.

As we describe above, the research database used in the present paper will provide opportunities for future studies on how comorbidities affect outcomes of first-line OA intervention and OA progression. Before doing that, it is important to determine the generalizability of the BOA population, to help interpretation of future study findings. This is why we would like to argue that the present study is needed. We have in this study, together with a previously published paper, made an effort to describe the OA population in BOA in comparison with that of the general Swedish population, from a health perspective. In the previous paper, we described the two populations from a socioeconomic perspective.[7] In contrast to previous research, the OA population in the BOA Register turned out to have an overall higher socioeconomic status (SES) than the general population. With the well-known association between low SES and poorer health, the present study thereby aimed to both describe comorbidities in this OA population, and to distinguish if detected socioeconomic differences could be explained by differences in comorbidities. Results from future outcome studies on the OA population in BOA will thus reflect the generalisability of the results from this present study on comorbidities, and in addition to our previous study on SES.

It is correct that the BOA Register only reaches a selected proportion of individuals who seeks health care for their OA problems. This discrepancy between treatment recommendations in guidelines, and the treatment that OA patients receive is a general problem among patients with OA. Studies have shown

that only about 50% of people with knee and/or hip OA receive treatment according to the guidelines.[8] Evaluation of the outcomes of the BOA intervention (education and exercise) [9] are in line with results from reviews on first-line OA exercise treatments [10, 11], and with other similar OA programs,[12] which indicates that this OA population reflects general OA populations at this stage of their disease (seeking healthcare and reference to first-line intervention).

We have also been careful in the present manuscript, to express only that our conclusions are drawn upon this specific OA population; "individuals referred to first-line intervention for hip and knee OA".

Action:

Regarding research of comorbidities in OA:

In the discussion section, page 21, we have now added the following sentence:

"Knowledge of how OA and comorbidities associate is still largely lacking and further research is needed.[5]"

Regarding generalisability;

We have now added the text marked with red in the discussion section, page 24:

"The results from the present study are applicable on OA individuals referred to first-line intervention, but are also in line with several previous studies,[2, 5, 13] describing comorbidities among individuals with OA compared with general populations."

Regarding exclusion criteria in the method section, paragraph Study design and participants at page 6;

We have now added the text in red to what was previously written in the method section, :

Patients with joint problems caused by another condition (e.g., sequelae of hip fractures, chronic widespread pain, inflammatory joint diseases or cancer), patients who have undergone total joint replacement within the previous 12 months or other surgery of the knee or hip joint within the previous 3 months do not meet the criteria for inclusion and are therefore excluded from registration in the BOA Register.

Since previous, this is also highlighted as a limitation in the discussion section, page 24: *Another limitation to bear in mind when interpreting the results from the present study is that the criteria for registration in the BOA Register exclude individuals who have other severe diseases*

Regarding OA in multiple joint;

Information on this has now been added to Table 1, at page 8.

3. The way how authors have dealt with comorbidity data is very problematic. It is great that they have used drug-data to supplement the often limited hospitalization data and the study also covers a large number of different conditions and is not limited to comorbidity indices. However, having OA and receiving care for it increases the odds for receiving care to other conditions as well and the comorbid conditions being registered. Table 2 shows that there are major differences in prevalences of different conditions. Hence, calculating prevalences of different types of comorbidities (CV disease, endocrine disease and so on) based on that any of registers have captured the conditions feels like counting apples with oranges. The results are currently dominated by the drug data (RxRisk) and there the problem is that the list of conditions includes many that could not be considered comorbidities (smoking cessation, pain, inflammation/pain). It is confusing that "with the Charlson Comorbidity Index, the OA population showed a slightly lower prevalence of comorbidities compared with the reference cohort" which opposes other results and the authors' conclusion. These limitations and their consequences have not been adequately in the discussion.

Reply:

Thank you for the possibility to explain our focus further.

Today there is no gold standard in what method to use when comorbidities are identified and studied. As we write in the discussion section, the ability to use indices is necessary when handling large datasets like this one. Even though some conditions may not be considered as comorbidities, they are included in the indices and therefore presented by us. All indices in the present study have also previously been used

in OA populations.[14, 15] As you write, comorbidities are identified from two different types of dataset, for both populations (the OA population and the reference cohort). We use drug-data and hospital data, which of course affects what outcome we get, since some diseases are easier detected with ICD-10 codes while others from drug data.

The fact that the Charlson Comorbidity Index identifies a slightly lower prevalence of comorbidities in the OA population is probably an effect of the exclusion criteria of the BOA Register, to exclude those with more severe diseases, which is the type of diseases that the Charlson mainly identifies.

Action:

We have now re-written the section in the discussion where we describe and reflect on the use of the indices to identify comorbidities, page 22-23.

In the limitation section (on page 23 and 24), we have also added the following reflections:

“To seek health care for a condition like OA may increase the probability to also be registered for other diagnoses, and/or receiving prescriptions of drugs for other conditions.”

and

“The contradictive results that the Charlson Comorbidity Index displayed a slightly higher prevalence of comorbidities in the reference cohort could also be an effect of the exclusion of individuals with severer diseases in the BOA Register, since the diseases detected by Charlson consistently are of a somewhat more serious nature.”

4. Some of the results are obvious consequences of inclusion criteria (like lower cancer in the BOA cohort) limiting the value of some of the findings. It is possible that some results that conflict with previous studies (depression, anxiety) are related to the methodology as well.

Reply:

Yes, we agree that the results are affected by the criteria for inclusion and exclusion, but also an effect of the intervention itself. Individuals with more comorbidities are often excluded from exercise treatments in daily clinical practice. To participate in first-line OA intervention with education and exercise demands an active and motivated patient, which can be a reason why diseases like depression and anxiety are not more common in this specific OA population. This is also why we argue that this paper is needed and why we aim to describe and clearly specify this selected OA population.

Action:

None

5. Obesity is a major confounding and/or explanatory factor in this study. However, it has been ignored and has not been discussed as limitation. To my understanding, BOA register includes BMI data but it was not used.

Reply:

This is a relevant point of view, and it is correct that the BOA Register includes data on BMI for this OA population, however, we have no access to the same data for the reference cohort.

Action:

We now included the following sentence in the section discussion, paragraph; Limitations and strengths at page 23:

“Obesity increases the risk of developing several chronic conditions including OA [16] and may be a confounding factor of the results. Since we only had access to data on body mass index for the OA population, but not for the reference cohort, we could not adjust for obesity.”

6. Even with the use of drug data, the prevalences of certain comorbidities is lower than one would expect (for example, diabetes). Partly this - and especially the lower prevalences with Elixhauser and Charlson indices - is likely related to the fact that comorbidities were registered only from three years before the index date. This selection is not reasoned in any way.

Action:

We have now re-written the section in the discussion where we reflect on the indices (page 22-23).

7. It is unclear how did the patients consent to participate. It feels unlikely that all would be willing to be registered to a quality register.

Reply:

Thank you for giving us the opportunity to clarify. Regarding the OA population collected from the BOA Register; at registration in the BOA Register, patients received oral and written information about the registration, that the data may be used for research and that they have the possibility to abstain from registration.

Regarding the reference cohort; informed consent was waived according to standard practice in Sweden, as no individuals in this population-based cohort used for analysis could be identified since the government agency Statistics Sweden replaces each PIN with a serial number to anonymize data before returning data to the researchers.

Furthermore, informed consent of participation is not required in register-based research according to Swedish law and ethical boards in Sweden. The role of informed consent in register-based research in Sweden has previously been described by Ludvigsson et al.[17]

Action:

We have now re-written and added information in the Method section, paragraph Ethical approval and trial registration (page 15-16) (new text is marked with red):

“The study was approved by the Regional Ethical Review Board in Gothenburg, Sweden (entry number 1059-16), and the study was registered at clinicaltrials.gov, with trial registration number NCT03438630. The study followed the legal and ethical frameworks of informed consent in register-based research, as described by Swedish law and ethical boards.[17] All patients in the BOA Register received oral and written information about their registration, including information that data may be used in research. Regarding the reference cohort, informed consent was waived according to standard practice in Sweden, since no individuals in this population-based cohort could be identified. Statistics Sweden replaces each PIN with a serial number to anonymize data before returning it to the researchers.”

Minor concerns (include but are not limited to the following):

- Abstract could include more data about differences in individual comorbidities as well as about the age effect.

Action:

We have now added data on age effect in the abstract, but are unfortunately limited by the author instructions maximum words to add more data.

- *Some of the references feel a bit old.*
- *Use of references is not fully adequate. For example, there are two studies about future projections of OA epidemiology that are used as references for text related to symptoms of OA in the introduction.*

Action:

We have worked through the references, updated to newer or more appropriate as suggested. Some older references are however, the seminal references for diagnosing OA and for the indices, and we believe are the most appropriate to justify some of our statements.

- *Differences in the results of the three different comorbidity tools should be discussed.*

Action:

Yes, we have now added this in the discussion. Please, see our response and action under comment number 3.

- *It is surprising that there are differences in baseline data like being born in Sweden and marital status although the selection criteria of reference population were not strict.*

Reply:

Yes, there are differences in SES (including mentioned socioeconomic indicators) between this OA population and their references, with this OA population displaying an overall higher SES. We have previously shown those differences in a publication.[7] The populations were matched on age, sex and residence (counties in Sweden), at date of inclusion in the BOA Register, not on any other variables.

Action:

None

Reviewer: 2

Dr. Gudmund Gronhaug, østfold hospital trust

Reply:

Thank you for taking the time to review our paper, your positive comments and thoughts regarding how to further improve the manuscript. We reflect and answer below, line by line.

I have a few comments to the authors that, in my opinion, need to be addressed.

I will present my concerns in the line of order as they appear in the manuscript.

Introduction

Page 6 Line 45 - 58 is presenting the benefits of first line interventions for OA patients. As this is not the aim of the study I am not sure if this is needed in the introduction.

I would like the authors to present more details on how and why OA patients are facing a possible downward spiral of comorbidities and loss of function and quality of life as a result of OA.

Reply:

Yes, we agree that the intervention is not directly in the aim of the study. However, we would like to argue that it should be kept in the introduction since it defines and explains the OA population that used in our study, which is important to later discuss generalisability and limitations of our results.

Action:

Regarding details of the effect of comorbidities in patients with OA, we have now added the following sentences to the introduction (page 5):

“An increased number of comorbidities has shown to be associated with a greater deterioration of OA regarding function and pain, but the temporal relationship and causality are yet to be determined.[18]”

Page 15 Indicators of socioeconomic status

Using education as a socioeconomic marker is fine. Still, when working with statistics of Sweden I suppose it also was possible to get data on status of employment for the whole study group and not just the

As OA patients are more often on sick leave and face threats of losing their job it would strengthen the study to use employment status as well as educational level as markers of SES.

Reply:

Yes, we agree. As you write, sick leave is common in OA patients. We have in a previously paper [7] studied the OA population from the present manuscript, from a socioeconomic perspective with a comparison with that of the general Swedish population. The results from that study showed that sick leave was more common, also in this OA population compared with their matched references. However, this OA population was at the same time employed to a higher degree than their references. They also had an overall higher SES than the references from the general population, indicating that this first-line intervention program may not reach more socioeconomic disadvantaged groups of individuals. Therefore, with the well-known connection between SES and health, one of the aims of this study was to examine if the detected differences in between the populations may be explained by differences in comorbidities.

Statistics Sweden deliver the socioeconomic data used in the previous and the present study. They collect it from several sources, such as the Swedish Tax Agency and the Swedish Social Insurance Agency, and then compiling it in the Longitudinal Integration Database for Health Insurance and Labour Market Studies (LISA), a database that contains detailed data on individual level. The LISA database is often used for health and socioeconomic research, but is often affected by delays in the collection of data from some sources and long process times. In our case, socioeconomic data on employment, income and family type was affected by delay so we do not have access to data for the year 2016 for those variables.

Action:

Please, see action regarding the use of employment status under the next comment.

Page 16 line 1-8

Statistical analysis of other SES markers

If employment did not affect the results different than the educational level it would be interesting to see the analysis and have them presented. Maybe as a sub group analyses for the group that is included before 2015?

Reply:

Please, see also the reply above, regarding why we do not have access to data after 2016.

Action:

We have now included income, employment and family type as socioeconomic factors in the model for individuals included before 2015, as a subgroup analyse (page 15). This is now presented in a supplementary file (Supplementary file 3.)

Line 13 - 27 Parts of this section belongs in the methods section and not the statistics section. Please check it.

Action:

We have now rewritten this part so it is better suited under just the statistics section (page 15), to simplify for the reader.

“Because nearly two-thirds of the OA population but only one-third of their references had prescriptions for anti-inflammatory medications (ATC codes M01A–M01H), drugs that increase the risk of developing gastrointestinal comorbidities,[19] and thereby often are prescribed preventive in combination with gastro-protective drugs (ATC codes A02BA01–A02BX05), we also chose to do separate analyses of those who were not using anti-inflammatory drugs to try to establish the real prevalence of gastroesophageal diseases.”

Line 43-44 This information is not about this study is it? It is OK not to involve patients in all studies.

Action:

We agree and have now removed those sentences from the manuscript (page 15, under the paragraph Patient and public involvement).

Page 20 separate analyses of subgroups

It would be interesting to see a full sub-group analysis of comorbidities in the group of this who have affected BOTH hip and knee compared with only knee or hip and their matched controls.

Reply:

We agree, however from the BOA Register we only have access to information on the joint most affected by OA (hip or knee) and if the problems were unilateral (class A on the Charnley score), bilateral (class B) or in multiple joint or if some other condition were affecting the patient's ability to walk (class C). Class C may thus also include individuals with other problems than OA, why we have chosen not to analyse those sub-groups.

Action:

We have now added descriptive information about the proportion in the OA population who was classified as B or C on Charnley score (please see Table 1 at page 8).

Discussion

Although a lot of the discussion is based on assumptions I find it convincing and well balanced based on the findings.

Still, in the limitations, or the discussion, it should be mentioned that it is likely that some of the controls probably also are OA patients. In my opinion this is likely to contribute to the results and should be mentioned.

Reply:

Thanks for your positive reflections regarding the discussion section in the manuscript. Regarding the possibility that there could be OA individuals also among the references, this is already mentioned as a limitation, on page 24, *“In addition, there could be individuals with OA in the reference cohort”*.

Action:

However, to further clarify this limitation, we have now added (marked with red):

“In addition, there could be individuals with OA in the reference cohort, which may affect the results”

Another more philosophical question that might not be appropriate to discuss in this paper is how to define comorbidities in the control group. For the OA population one extra disease is a comorbidity. For the controls the first disease is not a comorbidity as it is not linked to a baseline diagnosis. This is to some extent taken into account by using the comorbidity indexes as I understand it. But it might be worth mentioning how the authors are overcoming this philosophical obstacle of counting diagnoses.

Reply:

Yes, this is an interesting question and not entirely easy to address, in this context. After considering various definitions, we decided to use the definition of comorbidities, described by Valderas et al:[20] *“presence of additional diseases in relation to an index disease in one individual”*. However, as suggested by the reviewer, this definition may not be the most appropriate for the controls. A more correct concept for the reference cohort may had been to define them as having or not having multimorbidity, defined of Valderas et al as *“the co-occurrence of multiple chronic or acute diseases and medical conditions within one person”* without any reference to an index condition”. Since the manuscript needs to be stringent and easy to read, we chose not to switch between different concepts, but use the same for both population. This is the same way as it is treated in other studies were OA- and non-OA populations are compared.[5]

Action:

In the method section, the paragraph Identification of comorbidities (page 9), we have now added the following sentence:

“When describing the OA population and the reference cohort, we use the term comorbidities throughout the present study, defined as “presence of an additional disease in relation to an index disease in one individual”, even though a more correct definition regarding the reference cohort would be morbidity or multimorbidities”.

VERSION 2 – REVIEW

REVIEWER	Gronhaug, Gudmund østfold hospital trust, Department of physical medicine and rehabilitation
REVIEW RETURNED	14-Jun-2021

GENERAL COMMENTS	Thank you for a taking my concerns into consideration. I find my concerns and questions well answered and elaborated both in the manuscript, supplementary file and letter to the reviewers. Looking forward to follow your next publications from this vast database.
--